# 3D Printing of Steak-like Foods Based on Textured Soybean Protein

**DOI:** 10.3390/foods10092011

**Published:** 2021-08-26

**Authors:** Yangyang Chen, Min Zhang, Bhesh Bhandari

**Affiliations:** 1State Key Laboratory of Food Science and Technology, Jiangnan University, Wuxi 214122, China; giggle_chenyang@163.com; 2International Joint Laboratory on Food Safety, Jiangnan University, Wuxi 214122, China; 3Jiangsu Province Key Laboratory of Advanced Food Manufacturing Equipment and Technology, Jiangnan University, Wuxi 214122, China; 4School of Agriculture and Food Sciences, University of Queensland, Brisbane, QLD 4000, Australia; b.bhandari@uq.edu.au

**Keywords:** 3D printing, textured soybean protein, steak-like foods

## Abstract

Due to the lack of a sufficient amount of animal protein and the pursuit of health and reduced environmental impact, the global demand for plant protein is increasing. This study endeavors to using textured soybean protein (TSP) or drawing soy protein (DSP) as raw materials to produce steak-like foods through 3D printing technology. The textural difference between fried 3D printed samples and fried commercial chicken breast (control) was studied. The results show that different ink substrates (TSP and DSP) and hydrocolloids (xanthan gum, konjac gum, sodium alginate, guar gum, sodium carboxymethyl cellulose, and hydroxyethyl cellulose) were the keys to successful printing. The ink composed of TSP and xanthan gum had the best printing characteristics and sample integrity after frying. It was found that different infilling patterns and infill rates had a significant effect on the texture properties of the fried samples. When the triangle infilling pattern was used at an infill rate of 60%, the product had had the closest hardness (2585.13 ± 262.55), chewiness (1227.18 ± 133.00), and gumminess (1548.09 ± 157.82) to the control sample. This work proved the feasibility of using 3D printing based on plant protein to produce steak-like food with texture properties similar to chicken breast.

## 1. Introduction

Meat has always been welcomed by most consumers, not only because it has always been regarded as a source of high-quality protein, but also because of its unique taste and texture. However, excessive intake of meat products is unhealthy because meat contains cholesterol and polyunsaturated fatty acids, which can lead to cardiovascular disease [1]. In response, the World Health Organization has issued a recommendation for daily saturated fat intake of 15–30% and cholesterol intake of less than 300 mg/day [2]. Furthermore, with the growing population, people have also begun to pay attention to the adequacy of food. It is reported that a large number of people still suffer from malnutrition in underdeveloped countries, and malnutrition of protein and energy is also a major problem faced by most developing countries [3]. It was estimated that the global population will reach nine billion by 2050 [4], but meat production can only meet the needs of nearly eight billion people, and the meat industry needs to increase production by about 50–73% to maintain the daily demand of the growing population [5,6]. In addition to the problems caused by population growth, attention should also be paid to the sustainable development of the environment. Studies have shown that plant foods have lower greenhouse gas emissions than animal foods, with the largest (beef and mutton) emitting about 250 times more protein per gram than legumes [7]. Any reduction in the total number of animals raised for meat will better meet the requirements of environmental sustainability. In this environment, with the increasing awareness of healthy and sustainable food, many countries and regions are increasingly interested in plant-based meat substitutes [8]. Accordingly, the growth of the plant-based meat market is expected to increase from US $4.6 billion in 2018 to US $30.9 billion in 2026, and reach US $85 billion in 2030 [9]. As a kind of high-quality and cheap plant protein, soybean has been widely considered by consumers and researchers. Therefore, it has resulted in the production of some new soybean protein products, such as textured soybean protein (TSP) and drawing soy protein (DSP) [10,11,12]. These meat analogs should have the characteristics of texture, flavor, color, and nutrition of meat. It is not only a requirement of excellent new food but also a great challenge for modern food product development [13].

The unique bottom-up layer-by-layer stacking principle of 3D printing technology can print and produce complex fine structures that cannot be produced by traditional processing methods, which are widely used in a variety of industries, including military, automotive, textile, and some food industries [14]. Food 3D printing can combine 3DP and digital cooking technology to produce new types of foods with complex shapes, unique textures, and higher nutritional properties. In fact, 3D printing technology has been applied to food processing by its high-precision and unique processing methods [15]. For example, peanut protein and different fruit and vegetable powders are used to print foods of different colors and nutrients [16]; products developed for the elderly with swallowing difficulties, customized healthy snacks to meet the needs of children, and printed foods in various shapes or colors to meet the needs of consumers [17,18,19]. Due to the unique production method and high-precision characteristics of 3D printing, it can be possible to construct the fine structure of vegetable protein meat, achieving close to the texture of actual meat. 3D printing also allows customized protein meat shapes, flavors and colors, and even nutrients, meeting energy consumers’ need. Not only can 3D printing technology be used to narrow the sensory differences between plant protein meat and real meat, but plant protein meat can also be customized for the different preferences and needs of different consumers. Additionally, consumers are also very interested in 3D printing, a new food production technology [20,21]. Therefore, whether considering consumers’ attitudes towards 3D food printing, or the huge market prospect of plant protein meat, using 3D printing to make plant protein meat more realistic (in texture, color and smell, and even nutrition) is of great significance and demand.

Hydrogel is a kind of polymer with a three-dimensional grid structure, which has certain swelling and water absorption capabilities. It can enhance food texture properties by improving the elasticity, consistency, and stability of food [22]. In addition, hydrocolloids are widely used in 3D printing because the hydrogel is viscoplastic, and rapid photopolymerization is also conducive to high-fidelity printing of complex geometric shapes [23]. In previous studies, many hydrocolloids have been used in 3D food printing to improve the printability of printing inks or the texture properties of printed products. For example, xanthan gum has been shown to enhance the elasticity of polysaccharide gel ink, and it has also been shown to enhance the printability of pork and improve the texture of pork for those with difficulty in swallowing [24,25]; agar is proven to enhance the elasticity and viscoelasticity of edible gel materials [26]. In addition, hydrocolloids such as xanthan gum, guar gum, and cellulose derivatives have also been found to reduce oil absorption in fried products [27,28]. Therefore, this work discussed the effects of six hydrocolloids on the printability of TSP and DSP inks and determined xanthan gum and TSP inks for the production of steak-like foods. This work also explored the effects of different infilling patterns and infilling ratios on the texture of 3D printed products and determined a meat substitute that has no significant difference with the texture of the purchased chicken nuggets.

## 2. Materials and Methods

### 2.1. Materials

The materials and sources used in this work are shown in Table 1.

### 2.2. Preparation of 3D Printing Materials

In the process of 3D printing material formulation, TSP and DSP were dispersed in distilled water to fully soak and absorb water. The mixture was then beaten using a mixer (in the pre-test, compared the experimental method of crushing TSP and DSP before the mixture, and the printing effect was worse than that the method adopted in the experiment.). Next, inks with different components were prepared, and their composition is shown in Table 2. To prevent moisture loss of the prepared inks, they were wrapped tightly with a plastic cling film and stored in a refrigerator at 4 °C. Then, the samples were 3D printed and further analyzed.

### 2.3. Analysis of 3D Printing Results

In the process of measuring and analyzing the 3D printing effects, an extrusion-based food 3D printer (FOODBOT-MF, China Changxing Shiyin Technology Co., Ltd., Hangzhou, China) was used to print a preset pattern. The printer extruded the material by controlling the movement of the piston of the syringe cylinder through a nozzle according to the preset model. In the printing experiment, the main printing parameters were set as follows: the diameter of the syringe was 22 mm, the printing speed was 20 mm/s, the diameter of the nozzle was 0.8 mm, the height of the printing layer was 0.8 mm and printing temperature was 25 °C. A steak-like model (60 mm × 30 mm × 8 mm) was printed, which was stored in a disposable and covered square petri dish in a refrigerator at 4 °C.

### 2.4. Rheological Properties of Printing Materials

A rotary rheometer (Discovery HR-2, DHR, TA Instruments, New Castle, DE, USA) was used to determine the rheological properties of printing materials. The linear viscoelastic range of the sample was determined scanning at the constant frequency of 10 rad/s in the range of 0.01–10% using a parallel plate of 20 mm. The main parameters applied were as follows: the gap between the plates was 1000 μm, the experimental temperature was 25 °C, the shear rate ranged from 0.01 to 10.0 1/s, and the angular frequency ranged from 1 to 100 rad/s. It should be noted that when loading the sample, an excess sample on the edge was removed with a scraper and a layer of silicone oil was applied on the edge to reduce the water loss of the sample during the experiment. The above measurement experiments were repeated three times in each group, and then the average value was taken to draw the experimental curve.

### 2.5. LF-NMR Analysis

The moisture content and moisture distribution of materials are closely related to the structure and rheological properties of materials; in 3D printing, the rheological properties of printed materials are the key factors affecting the effect of 3D printing. Therefore, low-field nuclear magnetic resonance analyzer (MicroMR20-030V Muri, China Shanghai Niumag Co., Ltd., Shanghai, China) was used to determine the water distribution of the samples to be printed. The experimental method of Liu et al. [29] was adapted. About 3 g of sample was weighed, wrapped in food-grade thin cling film, put into a glass tube sample holder (diameter of 10 mm), and then the glass tube containing the samples was loaded into the LF-NMR analyzer. Finally, a series of Carr–Purcell–Meiom–Gill (CPMG) pulses were chosen to determine the relaxation time spin-spin (T_2_) of the sample. Each group of experiments was repeated three times, and then the average value was taken to draw the experimental curve.

### 2.6. Frying the Sample after Printing

Before frying, the printed samples were stored in a refrigerator at −80 °C for 10 min. After a short period of low-temperature freezing, the printed sample not only maintained the shape of the printed sample but also made the sample harder due to water crystallization, which is not easy to deform and damage. The frozen sample was heated in oil at 170 °C for 5 min, then removed and placed on baking paper. Afterward, it was cooled to room temperature and stored in food-grade sealed bags to prevent samples from absorbing excess moisture. An electronic vernier caliper (Shanghai Meinite Industrial Co., Ltd., Shanghai, China) was used to measure the size of the sample after frying, with the results accurate to one decimal place.

### 2.7. Texture Analysis

A texture analyzer (TA.XTC-18, Shanghai Baosheng Co., Ltd., Shanghai, China) was used to measure the textural properties of 3D printed samples after deep frying. The TA/36 probe was used to determine the texture properties of the sample (30 mm × 30 mm × 8 mm), applying the test speed of 1.0 mm/s, the probe movement speed (before and after the test) of 3.0 mm/s, and the trigger force of 5 g. The determination was repeated 10 times for each group of samples.

### 2.8. Moisture Measurement

Moisture content is an important component index of food, which will affect the quality, decay resistance, and texture properties of food. About 3 g of the fried sample was dried in an oven at 105 °C for 24 h to constant weight, transferred to the desiccator for cooling, then weighed and had its moisture percentage (*wt*/*wt*) calculated [30]. All samples were in triplicate.

### 2.9. Statistical Analysis

SPSS software (SPSS26; IBM SPSS Statistics, Chicago, IL, USA) was used for one-way ANOVA and Duncan tests at 95% confidence level for data analysis. All graphics in the experiment were drawn using Origin software (2020b, OriginLab, MA, USA).

## 3. Results and Discussion

### 3.1. 3D Printing Characteristics

3D printing based on extrusion can be divided into three stages: extrusion, recovery, and self-supporting [31]. These three stages require the ink to have the following conditions: (1) in the printing process, the ink can be squeezed out from the nozzle stably and uniformly; (2) the mechanical strength and viscosity can be quickly restored after extrusion; (3) after the ink is extruded and deposited, it should be self-supporting enough [15]. Therefore, in this work, the 3D printing properties of inks mixed with different plant proteins and hydrocolloids were studied. As shown in Table 3, the control group without hydrocolloid had poor printing performance for both proteins, where the DSP could not even be extruded and only the water that was not bound to the material was extruded. After adding different hydrocolloids, all inks can be extruded except for inks containing guar gum. Moreover, there are significant differences in the printing properties of different soy protein inks containing the same hydrocolloid. On the whole, the printing effect of TSP composite inks was significantly better than that of DSP inks with the same hydrocolloid. The samples showed that printed with TSP inks had better self-supporting performance and can basically maintain the shape of the pattern after printing. TSP inks except for the guar gum, and other hydrocolloid composite inks can be printed with preset patterns, with a certain level of self-supporting properties. In the case of samples printed with TSP ink, the self-supporting properties and the uniformity of the line were the best for added xanthan gum. Although the TSP ink containing sodium alginate or hydroxyethyl cellulose could be printed as per the preset pattern, it failed to print or can only print some lines at the skirt edge of the pattern. The TSP ink with sodium carboxymethyl cellulose could be printed as per the pattern and skirt completely, but the self-supporting properties of the sample after printing was worse than that of other inks, showing a certain degree of sample collapse. Therefore, a complete skirt edge could be printed only with the TSP ink with xanthan gum, and the printed sample had a certain shape maintenance ability. For DSP inks, only xanthan inks have better printing characteristics, and the printing lines are significantly better than other inks. Other DSP inks showed poor self-supporting properties after printing, because the samples collapsed and there was no complete and clear line, of which sodium alginate and sodium carboxymethyl cellulose inks showed the worst self-supporting properties.

The printing property of TSP ink was better than that of DSP ink, which may be due to different complex structures with hydrocolloids and binding with water. The composite printing ink with more closely bound water will be more uniform, can be better extruded, and will have a certain degree of self-supporting properties after extrusion. In contrast, the water that was not closely bound to the protein was subjected to extrusion pressure during the printing process, converting the semi-bound water to free water. Such samples have poor self-supporting properties, as shown by the deformation of the shape after printing. These results proved that the type of hydrocolloid and protein in the ink composition had an important effect on its printability, and the TSP ink with xanthan gum has been proved to be the ink with the best printing properties in this work.

### 3.2. Low-Field NMR Analysis of Inks

In 3D printing, the smooth extrusion of ink is the first condition for successful printing. It is closely related to its moisture distribution, and studies have shown that the relaxation exponent can also be used to judge whether the ink can be printed [32,33]. In LF-NMR analysis, the smaller magnitude of the transverse relaxation time (T_2_) proves the water closely bound to other non-aqueous components was low. In contrast, a large transverse relaxation time shows less closely bound water with the non-aqueous components in the sample with a higher degree of freedom of water [34]. As shown in Figure 1, inks composed of different proteins and hydrocolloids generally had three peaks: the T_21_ peak of the bound water with the relaxation time in the range of 0.1–10 ms, the T_22_ peak of the semi-bound water with the relaxation time in the range of 10–100 ms, and the T_23_ peak of the free water with the relaxation time in the range of 100–1000 ms [35]. It can be observed from Figure 1 that the water distribution in TSP and DSP inks was obviously different, but the water distribution trend of the same protein remained the same no matter what kind of hydrocolloid was added. It was found that although the hydrocolloid in the ink is also an important factor affecting the ink moisture distribution, its influence is weaker than that of the protein species. In other words, the use of a suitable protein ink matrix is the most important in this work. Because the relative degree of binding water in DSP ink was small, and the relative degree of free water was large, DSP inks had higher fluidity, so most of the samples printed by this ink showed poor self-supporting properties and printing accuracy. In fact, an excessive liquid property in printing inks has proven to be unsuitable for printing, due to its reduced printing geometric accuracy and self-supporting properties [36]. In Figure 1B, the T_22_ and T_23_ peaks of DSP without hydrocolloid were lower than those of ink with any hydrocolloid, which indicated that the water content of DSP ink without hydrocolloid was very little. Most of the water existed in a state that was not bound to the protein, which explains the phenomenon that DSP inks were only squeezed out of unbound water when printing. Combined with the printing results of DSP inks and the results of water distribution analyzed by LF-NMR, we can conclude that DSP inks are not suitable for printing in this work.

The T_22_ peak of TSP ink was located on the left side of DSP, which indicated that the binding degree of non-aqueous components of TSP ink to water was higher than that of DSP ink, and it has better printing accuracy and self-supporting properties. The printing results of different inks proved that TSP ink does have better printing characteristics than DSP inks. It can be observed from Figure 1A that the T_21_ peak of TSP ink and TSP ink with guar gum were lower than that of other inks, and the content of bound water was lower than that of other inks. They do not have suitable liquid properties and cannot be squeezed out smoothly and continuously, and the printing results of these two inks were also worst when determining the printing results. Additionally, the T_21_ peak value of TSP ink with sodium carboxymethyl cellulose was the largest, followed by TSP + hydroxy methylcellulose, TSP + xanthan gum and TSP + sodium alginate. This indicates that the water content of the bound water also decreases in order, so in the printing process, they are more and more easily squeezed out of the nozzle.

### 3.3. Rheological Properties of Different Inks

The rheological property of printing ink is an important indicator of whether 3D printing can be successfully printed according to the preset pattern and is closely related to the printing accuracy and printing results [37,38]. The ink suitable for printing should have the right viscosity (generally small) to ensure that the ink can be successfully extruded from the nozzle. However, the viscosity should not be too low, and the ink should have sufficient cohesiveness to connect other deposited layers to maintain the shape [18]. The storage modulus (*G*′) is generally used to measure the elastic behavior, which reflects the mechanical strength of the sample. The loss modulus (*G*″) is used to describe the viscoelastic behavior of the sample [39,40]. Figure 2 shows the storage modulus and loss modulus scanning results of TSP and DSP inks. Irrespective of protein type, the *G*′ value was higher than the *G*″ value (Figure 2A–D), showing that the ink materials were in an elastic dominant state [34]. Moreover, as the shear rate increases, the apparent viscosity showed a decreasing trend, which indicates that all protein inks are pseudoplastic fluids. However, the change of tan δ with the scanning frequency showed both an upward and downward trend, but the final value was less than 1, which indicated that the samples in the test showed solid-like behavior with a poor fluidity.

Figure 3 shows the viscosity curve of TSP and DSP inks. The TSP ink without added hydrocolloids in Figure 3A had the maximum viscosity, while the addition of hydrocolloid reduced the viscosity of TSP ink to a different degree, which all benefit from the shear-thinning behavior of the hydrocolloid. This shear-thinning behavior allows ink with high viscosity to flow effectively during printing so that it can be smoothly squeezed out from the nozzle [41]. The viscosity of the ink with xanthan gum is reduced the most, making the ink viscosity optimum. This is also consistent with the minimum loss modulus of the ink added with xanthan gum in Figure 2B and the minimum tan δ value of the ink containing xanthan gum in Figure 2E. These results showed that the TSP inks with hydrocolloids were more easily extruded from the printing nozzle, but the improvement varies among hydrocolloids. Among all hydrocolloids, the xanthan gum reduced the viscosity of TSP ink to the greatest extent and is most suitable for printing [29]. Figure 3B presents the viscosity curve of DSP ink: it can be seen that the viscosity curves of all DSP inks were inconsistent and show a very chaotic trend, which may be due to the lack of strong bonding between DSP and water, and DSP ink is not uniform. In fact, uneven DSP ink cannot be squeezed out during printing or intermittently squeezed out.

### 3.4. Post-Printing Frying of Samples

Having good printing characteristics is not the only criterion for choosing printing inks. The printed samples often need to be processed further before consumption. This work aimed to use 3D printing technology to print and produce fried meat-type chicken nuggets, so the printed samples need to be fried. The best printing ink should not only have good printing characteristics but also maintain the original shape during frying. Table 4 presents the result of the fried treatment of the printed samples. All treatment groups can be fried except the control and guar gum groups, which were not fried because of the poor printing effect. However, most of the groups did not maintain their original shape after frying, and some groups even spread out during the frying process and could not maintain any shape. This showed that hydrocolloids as hydrophilic polymers can enhance the water retention of water-containing food systems [42]. Additionally, the gelation of the hydrocolloid made the originally loose protein material form a more stable structure [43]. In addition, hydrocolloids can also reduce oil absorption during frying [27,44], which may reduce the intake of unnecessary oils in the diet [28,45]. Nevertheless, the effect of different hydrocolloids as gelling agents was different. Among the results of the frying process, the shape of the printed sample group added with xanthan gum maintained the best and had almost the same shape before frying. Although the printed samples with konjac gum and hydroxyethyl cellulose could maintain the shape overall, they were weaker than the xanthan gum group in terms of sample shape integrity, as some cracks or gaps on the sample surface were observed. The ink texture of the pure protein system of the control group was loose and could not be directly fried.

### 3.5. The Effect of Different Infilling Patterns on the Texture of Samples during Frying

The taste of food is a parameter that consumers give great importance; thus, it has a great impact on food’s acceptance by consumers. TPA analysis was used to evaluate the hardness, chewiness, and viscosity of samples [46]. The hardness of the sample reflects the force exerted by the compression of the sample to produce constant deformation, while adhesiveness refers to the force exerted from instability to stability of the semi-solid sample, which can characterize the chewiness of the sample and the acceptability of consumers [47]. Figure 4 shows the samples of different infilling patterns (grid, triangle, wiggle, honeycomb) when printing and during frying. The samples could be printed out completely with good printing characteristics in all infilling patterns and can retain their shape during the frying process. The cross-sectional view of the sample in Figure 4 shows that the internal structure of the sample differs for different infilling patterns. Except for the triangular infilling pattern which can effectively connect the interior and surface of the sample, different voids are formed on the surface and interior of the sample for other infilling methods. Table 5 presents the texture test results of samples with different infilling patterns. Control 1 was a purchased fried chicken product and control 2 was made from a model without 3D printing. It can be seen from the table that there were significant differences (*p* < 0.05) in hardness, springiness, chewiness, and gumminess between the 3D printed and purchased samples and the non-3D printed samples. There were also significant differences in texture properties between different infill patterns, which may be due to the different internal porosity of different infill patterns, and the action of force to break was also different [48]. Moreover, it can be observed from Table 5 that there was no significant difference (*p* > 0.05) in hardness, gumminess, and chewiness between the samples with triangular infill pattern and the target sample.

### 3.6. The Effect of Different Infill Ratios on the Texture of Samples after Frying

In printing, different infilling patterns and ratios can affect the printing accuracy and texture characteristics of 3D printed objects. Figure 5 presents 3D patterns with different infilling ratios (triangular filling) and fried samples. From this figure, it can be observed that the samples with too low infilling ratios (20%) ruptured on the surface and a partial bulge formed during deep-frying, which was also the reason for the largest change in the height of the sample. The rupture degree of 20% infilling samples was greater than other infilling samples, which may be since the low infilling ratio will have a larger gap; thus, more oil enters the sample. From the cross-sectional view of the sample in Figure 5, it can be found that as the filling ratio increases, the inner grid structure becomes denser. The internal grid structure of the sample filled with 20% is destroyed after frying and cannot maintain the shape before frying. This is because the low infilling rate makes the sample contact with more oil during frying and more thoroughly evaporates the water in the sample. This causes the mesh that originally connected the epidermis to break and eventually form a bulge. Samples with a larger infilling ratio can be preserved without destroying the internal grid structure after frying. However, at high infilling ratios (80% or 100%), although the internal grid structure still exists, the inside of the sample cannot be properly fried due to the excessive infilling and still has a high moisture content. This will cause the interaction force between the internal grid structure of the sample to become smaller, and the fried sample will have a smaller hardness. As the contact surface of water and oil increases, the evaporation of moisture becomes more intense, leading to surface rupture of the samples. The moisture measurement results in Table 6 also show that the moisture content of the 20% infilling after deep frying is significantly lower than the other groups, only 1.95%. When the infilling rate was above 40%, the samples kept a complete shape after deep-frying and the sample size has only a small change.

Table 6 shows that the texture properties of the fried samples with different infilling ratios are significantly different. Among all samples, the smallest difference with the texture result of control group 1 was found for the sample with an infill ratio of 60%, with no significant difference of the control group in terms of hardness and chewiness. This may be related to the similar moisture content of the two groups. Moreover, it can be found that the texture properties of samples with different filling ratios have some regularities after frying. It is not difficult to see that the hardness and chewiness of samples with a high infilling ratio (≥80%) and low infilling ratio (<40%) are significantly different. However, in the high infilling ratio (80%, 100%), hardness was not significantly different. Additionally, there was a significant difference in the moisture content of the samples with high and low infilling ratios after frying; however, there was no significant difference in the moisture content of the samples when both were at high infilling ratios. Figure 6 shows the correlation analysis results between different infilling ratios and the texture characteristics of the samples after frying. The results show that the infilling ratio of the sample has a positive correlation with the moisture content, chewiness, and adhesion of the sample after frying; it has a negative correlation with the hardness and springiness. There is also a certain degree of correlation between the moisture content of the sample and the hardness, chewiness, and adhesion. It is not difficult to understand that the water content is high, and the sample is not deep-fried enough, so it has less hardness and greater adhesion and chewiness. In contrast, a low filling ratio causes the sample to be over-fried, lose a lot of water, become harder, and lose chewiness and adhesion. This proved that different infill ratios had a significant effect on the texture of fried samples.

## 4. Conclusions

This research aims to use 3D printing to produce steak-like foods with TSP and DSP inks as a substitute for meat. In this study, the printing characteristics of ink formulation containing TSP, DSP, and different hydrocolloids and the integrity of samples during deep-frying were investigated and compared. The ink formulation with TSP and xanthan gum had the best printing characteristics and maintained the sample integrity. It was also found that both the infilling ratio and the infilling pattern had a significant effect on the texture of the sample after frying. When the infilling pattern was triangular with an infilling rate of 60%, it was closest to the texture properties of the control group in hardness (2585.13 ± 262.55), gumminess (1548.09 ± 157.82), and chewiness (1227.18 ± 133.00). The meat substitute industry has a broad prospective market, and soybean protein is also becoming accepted by more and more people. It is also likely to become a nutritious and functional product. Additionally, the customizability of 3D printing can be further utilized to produce steak-like foods that meet the nutritional or energy needs of different groups (gender, age), as well as to personalize flavor, taste and senses, producing healthy, nutritious, tasty and attractive 3D printed foods.

## Figures and Tables

**Figure 1 foods-10-02011-f001:**
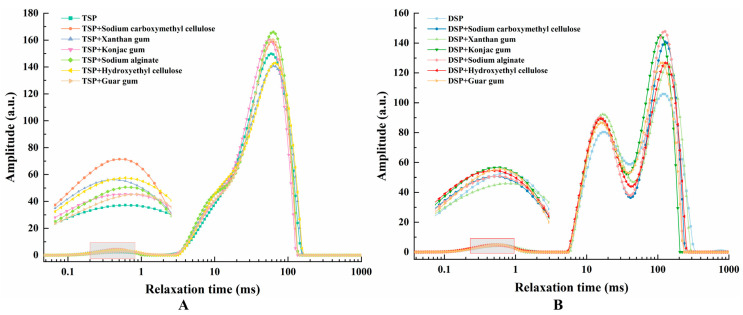
Moisture distribution of composite inks. (**A**) TSP added with different hydrocolloids. (**B**) DSP added with different hydrocolloids.

**Figure 2 foods-10-02011-f002:**
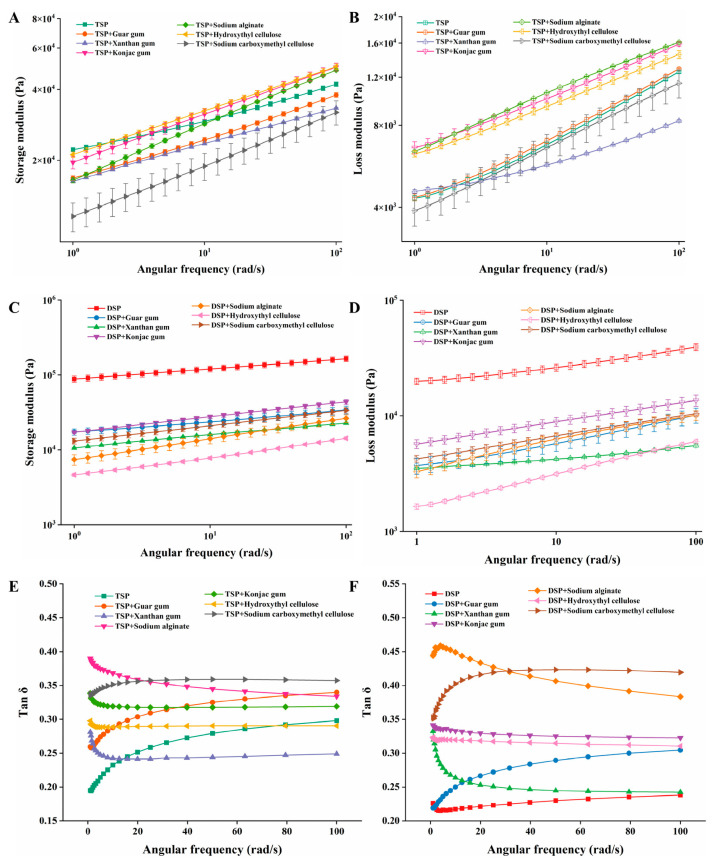
The rheological properties of different inks. (**A**,**C**) G′ of composite TSP and DSP inks added with different hydrocolloids. (**B**,**D**) G″ of composite TSP and DSP inks added with different hydrocolloids. (**E**,**F**) Tan δ of composite TSP and DSP inks added with different hydrocolloids.

**Figure 3 foods-10-02011-f003:**
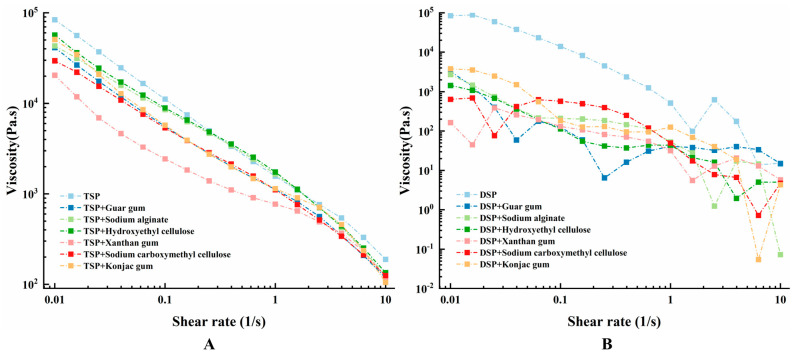
Viscosity curve of composite inks. (**A**) TSP with different hydrocolloids. (**B**) DSP with different hydrocolloids.

**Figure 4 foods-10-02011-f004:**
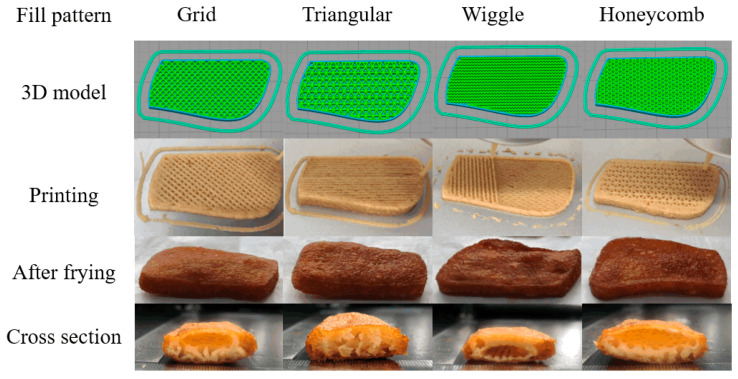
Fried 3D printed samples with different infill patterns (60% infill ratios).

**Figure 5 foods-10-02011-f005:**
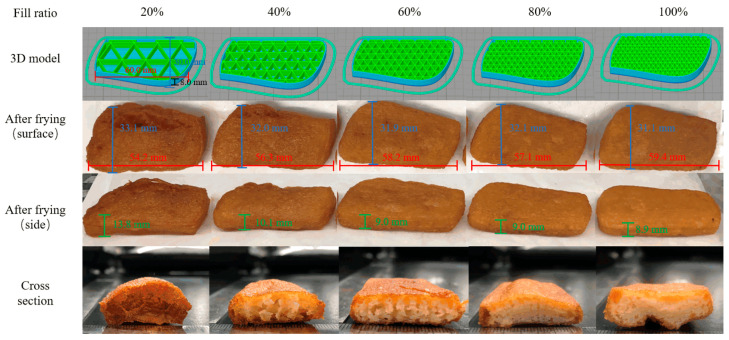
Fried of 3D printed samples with different (20–100%) infill ratios.

**Figure 6 foods-10-02011-f006:**
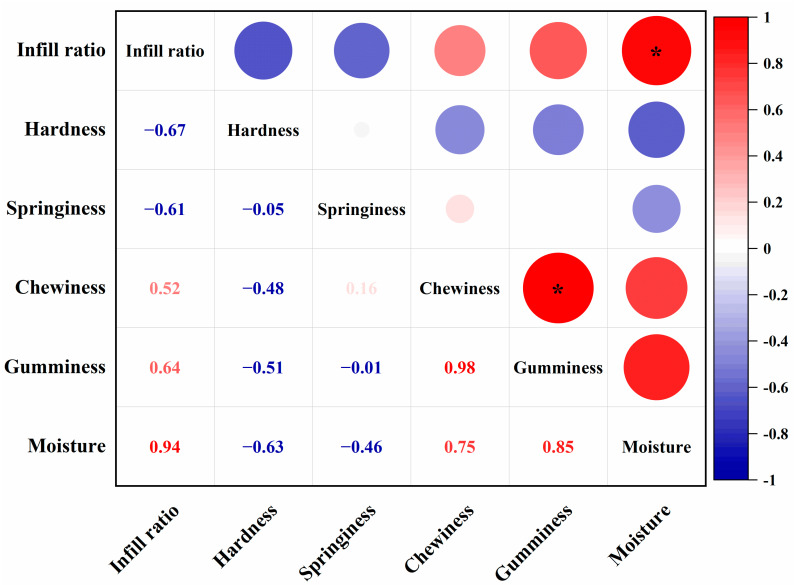
Pearson correlation analysis of the infilling ratio on the texture properties of 3D printed samples (* *p* < 0.05).

**Table 1 foods-10-02011-t001:** Materials and sources.

Material	Source	Grade
Textured soybean protein	Shandong Yuxin Biotechnology Co., Ltd. (Shandong, China)	Edible grade
Drawing soy protein	Anyang Detianli Food Co., Ltd. (Henan, China)	
Xanthan gum	Shandong Fufeng fermentation Co., Ltd. (Shandong, China)	
Potato starch	Nanjing Gan Juice Garden Sugar Industry Co., Ltd. (Nanjing, China)	
Konjac gum	Hubei Johnson Konjac Technology Co., Ltd. (Hubei, China)	
Guar gum	Shandong Yousuo Chemical Technology Co., Ltd. (Shandong, China)	
Agar	Shishi Huanqiu Qiongjiao Industry Co., Ltd. (Fujian, China)	
Isolated soy protein	Huiquan Biological Technology Co., Ltd. (Jiangxi, China)	
Sodium carboxymethyl cellulose	Henan Wanbang Industrial Co., Ltd. (Henan, China)	
Hydroxyethyl cellulose		
Sodium alginate		

**Table 2 foods-10-02011-t002:** The formula of different 3D printing inks.

Inks	Soy Protein Drawing (79.5%)	Textured Soybean Protein (79.5%)	Potato Starch	Agar	Isolated Soy Protein
Control	DSP	TSP	12.5%	2%	4%
Guar Gum (2%)	DSP + Guar Gum	TSP + Guar Gum	12.5%	2%	4%
Sodium Alginate (2%)	DSP + Sodium alginate	TS P+ Sodium alginate	12.5%	2%	4%
Xanthan Gum (2%)	DSP + Xanthan gum	TSP + Xanthan gum	12.5%	2%	4%
Konjac Gum (2%)	DSP + Konjac gum	TSP + Konjac gum	12.5%	2%	4%
Sodium Carboxymethyl cellulose (2%)	DSP + Sodium carboxymethyl cellulose	TSP + Sodium carboxymethyl cellulose	12.5%	2%	4%
Hydroxyethyl Cellulose (2%)	DSP + Hydroxyethyl cellulose	TSP + Hydroxyethyl cellulose	12.5%	2%	4%

**Table 3 foods-10-02011-t003:** Printing results of different protein and hydrocolloid composite ink systems.

Protein	Control	Guar Gum	Sodium Alginate	Hydroxyethyl Cellulose	Xanthan Gum	Sodium Carboxymethyl Cellulose	Konjac Gum
Textured Soybean Protein	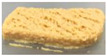	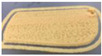	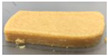	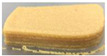	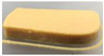	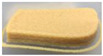	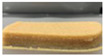
Drawing Soy Protein	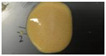	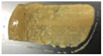	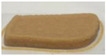	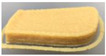	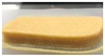	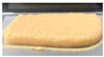	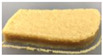

**Table 4 foods-10-02011-t004:** Fried samples performances of 3D printed TSP and DSP samples as a function of different added hydrocolloids.

Protein	Control 1 *	Control	GuarGum	SodiumAlginate	Hydroxyethyl Cellulose	XanthanGum	Sodium Carboxymethyl Cellulose	Konjac Gum
Textured Soybean Protein	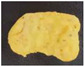	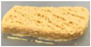 Not fried	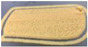 Not fried	Not shaped	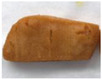	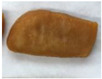	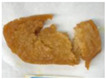	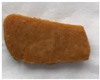
DrawingSoyProtein	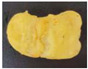	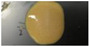 Not fried	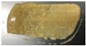 Not fried	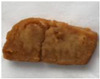	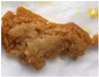	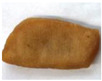	Notshaped	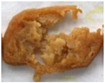

* Control 1 was a purchased fried chicken product.

**Table 5 foods-10-02011-t005:** Texture results of fried 3D printed samples with different infill patterns.

Infill Pattern	Control 1 *	Control 2 **	Grid	Triangular	Wiggle	Honeycomb
Hardness	2686.54 ± 203.71 ^a^	700.31 ± 90.69 ^c^	1781.06 ± 245.36 ^b^	2585.13 ± 262.55 ^a^	2461.17 ± 336.45 ^a^	2388.41 ± 310.77 ^a^
Springiness	0.84 ± 0.04 ^a^	0.76 ± 0.09 ^ab^	0.79 ± 0.02 ^ab^	0.79 ± 0.02 ^ab^	0.75 ± 0.05 ^ab^	0.80 ± 0.03 ^ab^
Chewiness	1341.25 ± 203.64 ^a^	400.95 ± 90.45 ^c^	920.47 ± 100.87 ^b^	1227.18 ± 133.00 ^a^	1192.92 ± 236.17 ^a^	1250.13 ± 155.19 ^a^
Gumminess	1554.60 ± 136.55 ^a^	489.64 ± 81.08 ^d^	1161.65 ± 151.16 ^c^	1548.09 ± 157.82 ^a^	1405.71 ± 116.84 ^ab^	1371.33 ± 124.39 ^b^

* Control 1 was a purchased fried chicken product. ** Control 2 was made from a model without 3D printing. The different superscript letters represent differences considered statistically significant (*p* < 0.05) for the textured properties of 3D printed samples after frying in different infill patterns.

**Table 6 foods-10-02011-t006:** Texture and moisture results of samples with different infill ratios after frying.

Infill Ratio	Control 1 *	20%	40%	60%	80%	100%
Hardness	2686.54 ± 203.71 ^b^	4220.00 ± 584.12 ^a^	1594.58 ± 286.47 ^c^	2585.13 ± 262.55 ^b^	1970.14 ± 605.56 ^c^	1778.70 ± 518.61 ^c^
Springiness	0.84 ± 0.04 ^a^	0.75 ± 0.12 ^bc^	0.81 ± 0.05 ^ab^	0.79 ± 0.02 ^abc^	0.73 ± 0.06 ^bc^	0.71 ± 0.03 ^c^
Chewiness	1341.25 ± 203.64 ^a^	376.12 ± 87.81 ^d^	689.12 ± 196.65 ^c^	1227.18 ± 133.00 ^a^	983.21 ± 176.40 ^b^	751.75 ± 87.40 ^c^
Gumminess	1554.60 ± 136.55 ^a^	533.59 ± 167.45 ^c^	863.36 ± 175.58 ^b^	1548.09 ± 157.82 ^a^	1365.02 ± 352.63 ^a^	1100.16 ± 194.50 ^b^
Moisture	45.88% ± 4.44% ^b^	1.95% ± 0.49% ^e^	15.76% ± 1.20% ^d^	42.61% ± 0.07% ^b^	50.12% ± 3.34% ^a^	51.11% ± 1.57% ^a^

* Control 1 was a purchased fried chicken product. The different superscript letters represent differences considered statistically significant (*p* < 0.05) for the textured properties and moisture content of 3D printed samples after frying in different infill ratios.

## Data Availability

The data presented in this study are available within the article.

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
