# Peer review of "3D Printing of Steak-like Foods Based on Textured Soybean Protein"

_foods, 2021, doi:10.3390/foods10092011_

Round 1

Reviewer 1 Report

The aim of the article was interesting but the discussion should be improved.

Some detailed  remarks

Chapter Preparation of 3D printing materials: It will be more clear if the composition of material is presented as percentage in Table. Also all variant should be included.  

Chapter Texture analysis What was the size of deformed sample. Three times of texture measurement are a very low number of reportions?

Chapter Moisture measurement It should clear mention that was the moisture measurement after printing or after frying?

How Authors estimate some mechanical parameters as Hardness, Chewiness Gumminess and Chewiness. I believe that it was not TPA which was not correct for cutting or shear testing of foods.

The main drawback of the article is poor discussion of the obtained data, The relation between of the obtained parameters should statistically analysed and discussed. The correlation test  or PCA analysis.

Author Response

Reviewer 1:

The aim of the article was interesting but the discussion should be improved.

Some detailed remarks

  1. Chapter Preparation of 3D printing materials: It will be more clear if the composition of material is presented as percentage in Table. Also all variant should be included.

Response: Thank you very much for your suggestions. As suggested, we have added a table describing the composition of the ink, including all variants. Please see the added Table 1. The formula of different 3D printing inks. And we also re-edited 2.2. Preparation of 3D printing materials.

  1. Chapter Texture analysis What was the size of deformed sample. Three times of texture measurement are a very low number of reportions?

Response: Thank you very much for the comments on our manuscript. As suggested, we have re-evaluated the texture properties of the samples, and the measurement results are shown in the revised Table 5 and Table 6. The measured sample size is 30 mmÍ30 mmÍ8 mm, and the measurement is repeated 10 times for each set of samples. Please see the revised chapter 2.7. Texture analysis.

  1. Chapter Moisture measurement It should clear mention that was the moisture measurement after printing or after frying?

Response: We are very sorry for our negligence. We measured the moisture content of the samples after frying and added an explanation in section 2.8 Moisture measurement. Please see the revised lines 179-180.

  1. How Authors estimate some mechanical parameters as Hardness, Chewiness Gumminess and Chewiness. I believe that it was not TPA which was not correct for cutting or shear testing of foods.

Response: Thank you very much for the comments on our manuscript. In the original manuscript, we used the shear test method commonly used in meat to evaluate the texture properties of the samples. As suggested, we have re-evaluated the texture properties of the samples, and the measurement results are shown in the revised Table 5 and Table 6. The measurement method is TPA analysis, and the probe used is TA/36, the test speed of 1.0 mm/s, the probe movement speed (before and after the test) of 1.0 mm/s, and the trigger force of 5 g, and the measurement is repeated 10 times for each set of samples. Moreover, 2.7. Texture analysis part has been revised. Please see the revised lines 170-176.

5.The main drawback of the article is poor discussion of the obtained data, The relation between of the obtained parameters should statistically analysed and discussed. The correlation test or PCA analysis.

Response: Thank you very much for your feedback and comments on our manuscript. According to the suggestion, we increased the correlation analysis between the infilling ratios and the texture characteristics of the sample after frying and increased the discussion. Please see the added Figure 6 and the discussion added in section 3.6. (Lines 399-410)

Reviewer 2 Report

This article, on the study of a meat substitute by vegetable proteins, is globally of good quality. However, the remarks below would help to improve it.

General remarks:

The model foods described in this study are composed of vegetable proteins, but also of a significant proportion of additives. What about the final protein content compared to conventional meat? Knowing that, in the case of a diet excluding animal proteins, a 15-20% higher consumption, in mass, of vegetable proteins is necessary. Elements concerning the bioavailability of the plant proteins used would have been appreciated in the introduction.

The introduction is too general in relation to the subject of the study. It lacks information about the texturing of foods with hydrocolloids which is the subject of your study. Please add references to this subject.

The question of personalizing the diet is a bit off-topic here because you are looking for a meat substitute.

It would be necessary to better define, upstream, the basic matrices: TSP and SDP. Information from the literature concerning their respective structures would be welcome (remark L88).

Specific remarks:

L51: Reverse the 2030 and 2026 data for a more logical order

L88: Please define the term "soybean drawing protein" precisely

L87-94: Paragraph 2.1 could be replaced by a table and an introductory sentence to avoid repetition.

L113: Please indicate the temperature of the sample during the extrusion phase, also 25°C?

L140-149: Is a freezing step necessary? Is the texture not solid enough for frying at room temperature? Does this call into question the conclusions of the study? That the printed product is not usable without a freezing phase?

L159: Please specify the heating time.

L168: Presence of an error in the subtitle.

L169-175: This paragraph could be simplified, presence of repetitions

L256: For figure 1, please indicate the units on the y-axis.

L314: This assertion would have deserved to be verified experimentally by an assay of lipids related to the total mass of the sample.

L340-342: From a nutritional point of view, is it desirable to have a deep-fried food? Do you have an idea of the amount of oil absorbed by the food? What is the final protein/fat ratio?

L351: For the figure 4, please indicate the method of measurement of the dimensions, how to measure to 1/10 of mm with a ruler?

A picture of the control would also have been desirable to realize its macroscopic structure compared to the other samples.

L369: It would have made more sense to switch paragraphs 3.5 and 3.6. First choose the infill pattern, then evaluate the filling rate. Please indicate if there is a reason for this presentation choice.

Author Response

General remarks:

  1. The model foods described in this study are composed of vegetable proteins, but also of a significant proportion of additives. What about the final protein content compared to conventional meat? Knowing that, in the case of a diet excluding animal proteins, a 15-20% higher consumption, in mass, of vegetable proteins is necessary. Elements concerning the bioavailability of the plant proteins used would have been appreciated in the introduction.

Response: Thank you very much for your comments on our manuscript. As suggested, we determined the protein content of the final sample and the control group (purchased fried chicken product). The protein content in the final sample and the control group are as follows, final sample (13.36±0.35 g/100 g) and control group (11.07±1.38 g/100 g). It can meet the requirement of a 15-20% increase in the quality of plant protein consumption in the case of a diet that does not include animal protein.

  1. The introduction is too general in relation to the subject of the study. It lacks information about the texturing of foods with hydrocolloids which is the subject of your study. Please add references to this subject.

Response: Thank you very much for your feedback and comments on our manuscript. We have reorganized the introduction part, added information about the texturing of foods with hydrocolloids, and provided references. Please see the revised introduction part, or added lines 92-110.

  1. The question of personalizing the diet is a bit off-topic here because you are looking for a meat substitute.

Response: Thank you very much for your suggestions. As suggested, we have removed the personalized diet description of the introduction part and reorganized the introduction part. Please see the revised introduction part.

  1. It would be necessary to better define, upstream, the basic matrices: TSP and SDP.

Response: Thank you very much for your suggestions. As suggested, we have added a table describing the composition of the ink, including all variants. Please see the added Table 1. The formula of different 3D printing inks. And we also re-edited 2.2. Preparation of 3D printing materials.

  1. Information from the literature concerning their respective structures would be welcome (remark L88).

Specific remarks:

L51: Reverse the 2030 and 2026 data for a more logical order

Response: Thank you very much for your suggestions. According to the suggestion, we have changed the sentence “And the growth of the plant-based meat market is expected to increase from US$4.6 billion in 2018 to US$85 billion in 2030, and reach US$30.9 billion in 2026.” to “And the growth of the plant-based meat market is expected to increase from US$4.6 billion in 2018 to US$30.9 billion in 2026, and reach US$85 billion in 2030.”. Please see the revised lines 60-62.

  1. L88: Please define the term "soybean drawing protein" precisely

Response: Thank you very much for the comments on our manuscript. “Soybean drawing protein” is also expressed in some literature. The references are as follows. But based on your suggestion, we have changed “soybean drawing protein” to “drawing soy protein”. And the information in the figure and table has been revised. Please see the revised lines 63-65.

“Soybean drawing protein” references:

“Liu, X.; Zhang, X.; Zhu, C.; Wang, J. Nutrition and health function of soybean drawing protein and its application in food. In Proceedings of the Proceedings of the 2017 6th International Conference on Energy, Environment and Sustainable Development, 2017; pp. 11-12.”

“Drawing soy protein” references:

“Wen-tian, S. Development of Emulate Smoking Ham Using Drawing Soy Protein. Food Research and Development 2012, 12.”

“Liu, W.; Jiang, Z.; Jia, D.; Zhang, J.; Yao, K. Effects of solid-state fermentation degree by Aspergillus oryzae on the properties of drawing soy protein. China Oils and Fats 2019, 44, 128-131.”

“Kurose, T.; Urman, K.; Otaigbe, J.U.; Lochhead, R.Y.; Thames, S.F. Effect of uniaxial drawing of soy protein isolate biopolymer film on structure and mechanical properties. Polymer Engineering & Science 2007, 47, 374-380”

  1. 7.L87-94: Paragraph 2.1 could be replaced by a table and an introductory sentence to avoid repetition.

Response: Thank you very much for your suggestions. According to the suggestion, we have revised the content of paragraph 2.1 and added Table 1. Please see the added Table 1, and revised line 114.

  1. L113: Please indicate the temperature of the sample during the extrusion phase, also 25°C?

Response: Thank you very much for the comments on our manuscript. The temperature of the sample extrusion phase is also 25℃. We have revised the description of the printing conditions in the 2.3 Analysis of 3D printing results section and revised it to “……the diameter of the syringe 22 mm, the printing speed 20 mm/s, the diameter of the nozzle 0.8 mm, the height of the printing layer is 0.8 mm, and printing temperature 25°C.”. Please see the revised lines 132-134.

  1. L140-149: Is a freezing step necessary? Is the texture not solid enough for frying at room temperature? Does this call into question the conclusions of the study? That the printed product is not usable without a freezing phase?

Response: Thank you very much for your feedback and comments on our manuscript. The freezing step before frying is not necessary. In the preliminary experiment, we compared the difference between the fried samples at room temperature and the fried samples after freezing and found that there was no significant difference between them. And samples at room temperature fried texture will not be affected, showing the same texture as frozen fried samples. The reasons why we choose to fry after freezing are as follows:

  1. Printed samples at room temperature are soft before frying and need to be transferred from the printing unit to the frying unit during frying operations.In this process, the printed sample will be transferred many times, there is a great chance to cause the deformation and damage of the sample.The frozen samples are hard and will not be damaged or deformed during the transfer process. This is crucial for the production of samples with high precision and complex shapes.
  2. In addition to the reasons for freezing operation in the production process of the above,we also consider the application of this 3D printing product in the food industry. In industrial production, products will experience several steps of production, storage, transportation, and sales. We take into account that this 3D printing product needs to remain in the shape that has just been produced after printing in the storage, transportation, and marketing phases, and that no deformation or damage will occur during these phases. However, the samples at room temperature are not hard enough before frying, which is easy to cause deformation and damage in these operations. Freezing treatment provides a hard texture for the samples, which avoids the occurrence of these problems and can effectively protect the quality of products.

  1. L159: Please specify the heating time.

Response: Thank you very much for your comments on our manuscript. During the moisture determination of the sample, we heat the sample at 105°C for 24 hours to a constant weight. We have revised section 2.8. Moisture measurement, please see the revised 2.8. Moisture measurement part.

  1. L168: Presence of an error in the subtitle.

Response: We apologize for our negligence in this error. We have revised the subtitle of section 2.9, and have changed “2.9. Data analysis” to “2.9. Statistical analysis”.

  1. L169-175: This paragraph could be simplified, presence of repetitions

Response: Thank you very much for your comments on our manuscript. We have simplified this part of the content, and we have revised “In the printing process, the ink first needs to be able to be extruded from the nozzle tip, and then the extruded ink needs to undergo viscosity recovery after undergoing shear force. Finally, the ink needs to have a good self-supporting property to maintain the shape after printing. Therefore, the ink needs to have the following conditions: (1) in the printing process, the ink can be extruded stably and uniformly; ……” to “3D printing based on extrusion can be divided into three stages: extrusion, recovery and self-supporting. These three stages require the ink to have the following conditions: (1) in the printing process, the ink can be squeezed out from the nozzle stably and uniformly; ……”. Please see the revised lines 190-193.

  1. L256: For figure 1, please indicate the units on the y-axis.

Response: Thank you very much for your suggestions. As suggested, we have added the unit of the y-axis and modified Figure 1. Please see the revised Figure 1.

  1. L314: This assertion would have deserved to be verified experimentally by an assay of lipids related to the total mass of the sample.

Response: Thank you very much for your comments on our manuscript. In this work, we first selected the ink with the best printing properties based on printability and rheological properties. Then the samples printed with the best ink were fried, and the texture closest to the control sample (purchased chicken nuggets) was achieved through different infilling patterns and infilling ratios. We determined the fat content of the best sample and the control sample. The results of the determination were: the best sample (17.14±2.39 g / 100 g) and the control (26.63±1.94 g / 100 g). The results show that the lipid content of the 3D printed samples after frying is significantly lower than the control sample, which proves that the addition of hydrocolloids can reduce the oil absorption during frying. And this conclusion has been confirmed in some studies, and some references can be given.

  1. “Akdeniz, N.; Sahin, S.; Sumnu, G. Functionality of batters containing different gums for deep-fat frying of carrot slices. J Food Eng 2006, 75, 522-526, doi:https://doi.org/10.1016/j.jfoodeng.2005.04.035.”
  2. “Garmakhany, A.D.; Mirzaei, H.O.; Nejad, M.K.; Maghsudlo, Y. Study of oil uptake and some quality attributes of potato chips affected by hydrocolloids. Eur J Lipid Sci Tech 2008, 110, 1045-1049, doi:https://doi.org/10.1002/ejlt.200700255.”
  3. “Dogan, S.F.; Sahin, S.; Sumnu, G. Effects of batters containing different protein types on the quality of deep-fat-fried chicken nuggets. Eur Food Res Technol 2005, 220, 502-508, doi:10.1007/s00217-004-1099-7.”
  4. “Katzbauer, B. Properties and applications of xanthan gum. Polym Degrad Stabil 1998, 59, 81-84, doi:https://doi.org/10.1016/S0141-3910(97)00180-8.”
  5. “Sahin, S.; Sumnu, G.; Altunakar, B. Effects of batters containing different gum types on the quality of deep-fat fried chicken nuggets. J Sci Food Agr 2005, 85, 2375-2379, doi:https://doi.org/10.1002/jsfa.2258.”

  1. L340-342: From a nutritional point of view, is it desirable to have a deep-fried food? Do you have an idea of the amount of oil absorbed by the food? What is the final protein/fat ratio?

Response: Thank you very much for your feedback and comments on our manuscript. We measured the fat and protein content of the 3D printed samples and control samples after frying. The fat content is the 3D printed sample (17.14±2.39 g / 100 g) and the control (26.63±1.94 g / 100 g). The protein content is the 3D printed sample (13.36±0.35 g/100 g) and the control group (11.07±1.38 g/100 g). The final protein/fat ratio is the 3D printed sample (0.78) and the control (0.42). Furthermore, the purpose of this work is to use 3D printing to produce meat substitutes with a texture similar to steak-like foods. The control sample used in this work is the deep-fried chicken nuggets sold in the market, so we performed the same frying treatment on the 3D printed samples. And fried processed foods may be unhealthy, but they are popular with consumers because of their unique flavors. In addition, frying is only one of the processing methods chosen in this work, if the pursuit of non-fried food, can be achieved through other alternative frying processing methods, such as air frying. Air frying has been proven to be feasible to achieve a characteristic color and flavor similar to traditional frying products with little or no oil treatment. Further, air frying has been applied to 3D printed food. For example, Feng et al. (2020) used air frying to produce 3D-printed yam snacks, and Liu et al. (2020) used air frying to produce 3D-printed potato snacks with low oil content. Some references are as follows:

  1. “Feng, C.; Zhang, M.; Bhandari, B.; Ye, Y. Use of potato processing by-product: Effects on the 3D printing characteristics of the yam and the texture of air-fried yam snacks. LWT 2020, 125, 109265, doi:https://doi.org/10.1016/j.lwt.2020.109265.”
  2. “Liu, Z.; Dick, A.; Prakash, S.; Bhandari, B.; Zhang, M. Texture Modification of 3D Printed Air-Fried Potato Snack by Varying Its Internal Structure with the Potential to Reduce Oil Content. Food Bioproc Tech 2020, 13, 564-576, doi:10.1007/s11947-020-02408-x.”

  1. L351: For the figure 4, please indicate the method of measurement of the dimensions, how to measure to 1/10 of mm with a ruler?

Response: We are very sorry for our negligence. We used an electronic vernier caliper (Shanghai Meinite Industrial Co., Ltd.) to measure the size of the sample, and the results of one decimal place. We have modified the sentence “And a ruler was used to measure the size of the sample after frying.” to “And an electronic vernier caliper (Shanghai Meinite Industrial Co., Ltd.) was used to measure the size of the sample after frying, the results of one decimal place”. Please see the revised lines 167-169.

  1. A picture of the control would also have been desirable to realize its macroscopic structure compared to the other samples.

Response: Thank you very much for your suggestions. As suggested, we have added the macrostructure photos of the control group (purchased fried chicken product) to Table 4. Please see the revised Table 4.

  1. L369: It would have made more sense to switch paragraphs 3.5 and 3.6. First choose the infill pattern, then evaluate the filling rate. Please indicate if there is a reason for this presentation choice.

Response: Thank you very much for your feedback and comments on our manuscript. We have adjusted the contents of paragraphs 3.5 and 3.6, first discussed the influence of the filling pattern on the texture of the sample, and then discussed the influence of the filling rate on the texture of the sample. Please see the revised part “3.5. The effect of different infilling patterns on the texture of samples during frying” and “3.6. The effect of different infill ratios on the texture of samples after frying”.

Round 2

Reviewer 1 Report

The paper was improved and I can accept it.